# Cyberattack Path Generation and Prioritisation for Securing Healthcare Systems

Shareeful Islam [1,*,†] , Spyridon Papastergiou [2,†], Eleni-Maria Kalogeraki [2,†] and Kitty Kioskli [3,4]

1   School of Computing and Information Science, Anglia Ruskin University, Cambridge CB1 1PT, UK
2   Department of Informatics, University of Piraeus, 185 34 Piraeus, Greece; paps@unipi.gr (S.P.);
    elmaklg@unipi.gr (E.-M.K.)
3   Institute of Analytics and Data Science (IADS), School of Computer Science and Electronic Engineering,
    University of Essex, Essex CO4 3SQ, UK; kitty.kioskli@essex.ac.uk
4   Trustilio B.V., 1017 HL Amsterdam, The Netherlands
*   Correspondence: shareeful.islam@aru.ac.uk
†   Focal point, Belgium.

**Abstract:** Cyberattacks in the healthcare sector are constantly increasing due to the increased usage of information technology in modern healthcare and the benefits of acquiring a patient healthcare record. Attack path discovery provides useful information to identify the possible paths that potential attackers might follow for a successful attack. By identifying the necessary paths, the mitigation of potential attacks becomes more effective in a proactive manner. Recently, there have been several works that focus on cyberattack path discovery in various sectors, mainly on critical infrastructure. However, there is a lack of focus on the vulnerability, exploitability and target user profile for the attack path generation. This is important for healthcare systems where users commonly have a lack of awareness and knowledge about the overall IT infrastructure. This paper presents a novel methodology for the cyberattack path discovery that is used to identify and analyse the possible attack paths and prioritise the ones that require immediate attention to ensure security within the healthcare ecosystem. The proposed methodology follows the existing published vulnerabilities from common vulnerabilities and exposures. It adopts the common vulnerability scoring system so that base metrics and exploitability features can be used to determine and prioritise the possible attack paths based on the threat actor capability, asset dependency and target user profile and evidence of indicator of compromise. The work includes a real example from the healthcare use case to demonstrate the methodology used for the attack path generation. The result from the studied context, which processes big data from healthcare applications, shows that the uses of various parameters such as CVSS metrics, threat actor profile, and Indicator of Compromise allow us to generate realistic attack paths. This certainly supports the healthcare practitioners in identifying the controls that are required to secure the overall healthcare ecosystem.

**Keywords:** healthcare ecosystem; medical devices; cyberattack path; vulnerability; exploitability

## 1. Introduction

The healthcare sector is becoming more digitally connected due to the advancement of technology, and so the potential risk of a cyber incident will increase. The connectivity of medical devices with other software and information communication technology (ICT) infrastructures poses potential risks. There is increasing concern that the connectivity of these medical devices will directly affect healthcare service delivery and patient safety, which is unique compared to traditional computing systems [1]. Research studies have shown that the number of hacking incidents reported in healthcare was 42% more in 2020 [2]. The healthcare information infrastructure is equipped with medical devices that require both physical and cyber interaction, which can create new attacker capabilities [3]. It is necessary to identify the possible attacks and related paths that can pose potential risks

within the healthcare context. To our knowledge, this is the first study that focuses on the vulnerabilities related to healthcare devices and their dependencies on other information technology (IT) infrastructures to propagate an attack.

The present paper illustrates an evidence-based attack path discovery method, considering the unique characteristics of the healthcare information infrastructure, such as assets, and its cyber and physical dependencies, vulnerabilities, threat actor and user profile, and indicator of compromise (IoC). There are three main contributions of this work. Firstly, the proposed approach includes a systematic process for attack path identification based on the assets, dependencies, and vulnerabilities. It adopts the existing standards, such as common vulnerabilities and exposures (CVE) and the common vulnerability scoring system (CVSS), to identify and analyse the vulnerabilities relating to the attack paths. These repositories contain huge amounts of data about the vulnerabilities. The identified attack paths are prioritised based on the IoC, which shows the evidence of any attack. Secondly, a knowledge base is developed that consists of rule-based reasoning to identify the possible attack paths. The rules are based on certain conditions that are necessary for a successful attack campaign. This allows us to determine possible attack paths so that appropriate control actions can be taken for securing the system. Finally, a real healthcare use case scenario that processes big data of healthcare applications is considered to validate the applicability of the proposed method. The results show that it is a practical approach for the attack path generation and determine the necessary areas that need adequate protection for the overall cyber security improvement.

## 2. Related Work

There are a number of studies that focus on attack path discovery. Existing research treats attack path discovery as an important stage focused on identifying and understanding the routes in a network that potential attackers might follow to gain unauthorized access to a system. This section provides an overview of the existing related work.

The attackers first infiltrate vulnerable hosts to access the system and use the previous attack result as a precondition and repeat this process until they achieve the level of control desired. Previous studies [4,5] aimed to evaluate all possible attack paths in a network and to predict future attacks by combining components from a collaborative filtering recommender systems and attack path discovery approaches using Naïve Bayes and random forest. This method searches for all non-circular attack paths that exist between assets that belong to the network and induces a model where an attacker can gain access to information system sources following a directed path. The security weaknesses of an asset follow the vulnerability assessment, conducting a thorough analysis of the existing and potential threat landscape within a network that can be valued. A stochastic analysis is considered for the evaluation of cyberattack paths through sophisticated methods to measure the probability and acceptability of faults [6]. The development of threat scenarios can delineate the underlined threat landscape and thus facilitate the threat knowledge and improve the visualization [7]. Other groups consider Attack Trees or Attack Graphs, which are widely used approaches for considering threat analysis during the risk assessment process. The attack graph network measurement can be classified into structure and probability-based metrices to quantify network security [8] to illustrate the network's agility in taking preemptive measures to respond to attacks and stochastic-based metrices to estimate large nodes of networks. Another work focuses on attack modelling as a useful tool in risk assessment for cyber physical systems based on the attack vector within the technical and operational environment [9]. Attack graphs are considered a series of exploitation of atomic attacks, which can drive the process to an undesirable state and are used for various applications including threat detection and forensic analysis [10].

More recently, research has appeared that focuses on discovering and analysing attack paths using threat intelligence and vulnerability exploitation. The exploration of attacks based on threat intelligence data is collected using cloud-based web service in [11]. The attack surface classification methodology of mobile malware with known threat actors

through automated tactics, techniques and procedures (TTP) and IoC analysis is described in [12]. Hence, IoC is considered as a key parameter to analyse the attack surface and attackers' motivations. A further study [13] focuses on just certain parts of the network to identify and generate attack graphs. For instance, in this strategy, they assume that there is a privilege over an asset across the network. If it is accurate, that means that the user gained access to the asset. An attack path discovery in the dynamic supply chain is proposed using the MITIGATE method in [14]. The approach considers a dynamic risk management system to detect the vulnerabilities that can deliver attack paths based on certain criteria. It considers attacker capability, attack path and its length, and knowledge base for analysing the attack paths. Another work proposes a recommended system that focuses on possible methods that can be used to classify future cyberattacks in terms of risk management [15]. This approach considers the exploitability features for attack path generation and uses a multi-level collaborative filtering method to predict the future attacks. Another indicative example for supply chain context is presented by [16], where cyber threat intelligence is integrated into the cyber supply chain for analysing the threats and determining suitable control strategies. An integrated cyber security risk management integrates vulnerability and threat profile for risk management and predication [17]. In particular, various threat actor parameters such as skill, motivation, location and resources are considered important for determining the likelihood of the risks related to a specific threat. A distributed approach for attack path generation based on a multi-agent system is considered by [18]. It follows an in-depth search, where the performance is improved with the use of agents after a specific graph size.

The contributions presented above have greatly contributed to the identification and analysis of attack paths. Several observations have been made based on the existing literature. Firstly, there is a lack of focus on specific vulnerabilities and their exploitability that contribute to the attack path discovery, particularly in the healthcare sector. Additionally, there is also a need to understand the threat actors' profiles in terms of attacker capability and motivation, as well as target user profiles for a successful attack campaign. Healthcare systems consist of interconnected cyber systems and infrastructures at the physical and cyber levels for critical healthcare service delivery [16]. There is a pressing need to understand the possible attack paths and prioritise the paths so that possible control actions can be identified to ensure the security and resilience of healthcare service delivery. The proposed work contributes towards this direction and adopts the widely used CVE vulnerability database and CVSS scoring system to examine the vulnerabilities that exist within the healthcare system.

## 3. The Proposed Attack Path Discovery

A cyberattack path determines the possible routes that an attacker can propagate to execute an attack. In general, all high-impact cyberattacks have several phases where an attacker conducts lateral movement from the initial point to the target landing point. A healthcare ecosystem by its inherent nature is complex and interconnects with a number of medical and IT assets for service delivery. The attack within the ecosystem can propagate from any initial point to the final target asset depending on the attacker profile and motivation. It is necessary to understand the vulnerabilities within the attack surface to adopt suitable control measures. The proposed method follows the existing attack path discovery methods such as MITIGATE [17], cyber-physical attack paths against critical systems [18], and attack path discovery in a dynamic supply chain context [19] and extends with new parameters and rule sets to formulate the attack paths. Additionally, the proposed method adopts the widely used CVE vulnerability database and CVSS scoring system to examine the vulnerabilities that exist within the healthcare system [20–22]. There is also a need to understand the threat actor profile in terms of attacker capability and motivation for a successful attack campaign. The proposed methodology also adopts the NIST's SP800-30 guideline [23] for profiling the attacker. This section presents an overview of the proposed methodology in terms of the general assumptions and process.

### 3.1. Assumptions

The proposed attack path discovery method considers the following assumptions:

- The assets within the healthcare ecosystem are dependent upon each other for the healthcare service delivery;
- Each asset may link with single or multiple confirmed vulnerabilities published by the National Vulnerability Database (NVD) or CVE, which are required to be considered for the attack path generation. CVE contains a huge list of published vulnerabilities that assist in determining vulnerabilities related to specific healthcare assets;
- The threat actor needs a certain profile in terms of attacker capability (knowledge and skill) and access vector (local, adjacent, network, and physical) to exploit a vulnerability and discover an attack path;
- Each user within the healthcare system performs certain functionalities based on the roles and responsibilities. Threat actors could take advantage of target user profiles to execute an attack;
- Each attack path includes several variables, such as entry point asset, intermediate point (if any), target point asset, dependencies among the assets, and underlying characteristics of the vulnerability within the assets;
- The methodology follows the CVSS for attack path generation and vulnerability estimation. It mainly considers the base score metric values for generating the attack path.

### 3.2. Cyberattack Path Generation and Analysis Process

This section presents the attack path generation and analysis process, which consists of seven distinct steps. Each step performs specific functionalities and contributes towards the attack path discovery. It initiates source and target asset identification, followed by the vulnerability chain for a successful attack campaign. An overview of the steps is given below.

**Step 1**—Identify possible entry points: This initial step identifies the healthcare ecosystem's potential assets that the attacker may consider as an entry point to execute an attack. This can be a medical device and software that runs the device, hardware, or other assets within an ICT infrastructure. A medical device defined by the FDA as software, electronic and electrical hardware, including wireless, is a critical asset for healthcare systems. The entry point is a point of failure where the attack exploits the vulnerability to propagate the target point. Generally, the attacker spends a lot of time trying to understand the existing system, and specifically, the healthcare system consists of several interconnected healthcare and IT devices. Vulnerabilities within these assets can be exploited by an attacker to achieve their intention. Almost every aspect of the network and application has a potential entry point, and securing the weakest link principle should be followed by the healthcare entity in order to make it difficult for an attacker to identify the entry point.

**Step 2**—Determine asset dependencies: Once the entry point is identified, it is necessary to determine the dependencies of these assets within the healthcare ecosystem. The goal is to focus on the potential cyber interaction of the entry point asset. A cyber dependency of assets is assumed to be a cyber-asset pair (node) interrelation and/or interconnection (edge) aiming to fulfil a healthcare service delivery or specific operation over communication networks. For instance, it is necessary to exchange patient treatment data from various sources for clinical decision making. Such dependency is critical for an attacker to propagate an attack from the entry point to the target point.

**Step 3**—Identify possible target points: This step aims to identify the possible target points that an attacker strives to reach by following the entry point asset and associated dependencies. An attacker needs to exploit single or multiple vulnerabilities to reach target points to achieve its objective. This includes assets that are compromised at interim stages of an attack campaign. Therefore, the accessibility of a target point depends on the entry point asset, cyber dependencies, and capability of the actor for possible exploitation. This step develops assets' dependency graph that shows assets and their dependencies.

**Step 4**—Determine entry and target point vulnerabilities: Once the entry and target points are identified, it is necessary to determine the vulnerabilities related to the assets. These vulnerabilities are preconditions based on the attacker's profile for a successful attack path campaign. The goal of this step is to accurately reflect the exploitability level of the identified vulnerabilities and link the vulnerabilities to formulate the vulnerability chain. These vulnerabilities are identified by following the CVE and NVD published vulnerabilities entries. The proposed method follows a rule-based reasoning approach (filters) to produce the chain of sequential vulnerabilities on different assets that arise from consequential multi-step attacks, initiated from the entry points in order to exploit the vulnerabilities. Individual vulnerability is measured by following the CVSS base metrics and possible further exploitation for the vulnerability chain. CVSS allows us to determine the criteria relating to discoverability, exploitability, and reproducibility to materialize a threat relating to the vulnerability. Therefore, the vulnerability chain demonstrates and escalates the attack vector from local to network access or vice versa.

**Step 5**—Define the threat actor and user profile: A threat actor needs a certain profile to exploit a vulnerability for a successful attack. Depending on the asset dependencies and nature of the vulnerability, the profile may vary. The threat actor profile includes sub-attributes relating to attacker capability (very low, low, moderate, high, and very high) and location (local, adjacent, network, and physical) for an attack campaign by following the NIST SP800-30 guidelines. Additionally, it is also necessary to consider existing users of the healthcare ecosystem and their profiling, which may assist the threat actor to exploit the vulnerability. Depending on the user role and access rights to various systems and other assets, the user profiling can be categorized into three scales of high, medium, and low.

**Step 6**—Generate attack paths: This step of the described methodology aims to generate the possible attack paths against target point assets. The individual and chain vulnerabilities are examined using a number of parameters, including assets, vulnerabilities, threat actor profiles, and exploitability level, for this purpose. The vulnerability chain demonstrates a series of exploitation of vulnerabilities using appropriate access vectors and escalation of the privilege. We follow the individual and propagated vulnerability level to determine the attack path. This step is iterative to generate the possible attack paths for the chosen healthcare context and the impact of the vulnerability are considered for selecting the appropriate ones.

**Step 7**—Generate and prioritise an evidence-based vulnerability chain: Once the attack paths are identified, it is necessary to generate the vulnerability chains whose exploitation can lead to possible attack paths on given cyber-dependent assets. It is also necessary to collect the evidence relating to the attack path so that we can prioritise which paths need to be taken into consideration for suitable control measures. This step consists of a number of sub-steps (7.1–7.4):

**Step 7.1**—Identify the vulnerability chain: The attack path discovery relies on unique characteristics, i.e., entry and target point assets and related vulnerabilities, the threat actor's capability, and asset interdependencies to identify all possible paths that can be exploited to gain access by generating vulnerability chains. At this stage, only the vulnerability chains that are under the attack capability for the exploitation are considered.

**Step 7.2**—Assess the vulnerability chain: Once all vulnerability chains are identified, it is necessary to assess the vulnerabilities for a given chain. The step considers individual, cumulative and propagation vulnerability values:

The individual vulnerability assessment (IVL): This measures the probability that a threat actor can successfully reach and exploit a specific confirmed vulnerability in a given asset. We follow the Exploit Prediction Scoring System (EPSS) of individual vulnerability and CVSS 3.1 score metrics (if EPSS is not available) to estimate the vulnerability level. Hence, several external sources such CVSS, EPSS, exploit-db are considered for IVL. Table 1 shows the individual vulnerability assessment scales. A list of generic assumptions for calculating the probability is made.

**Assumption 1.** *If exploitability features (proof-of-concept exploit code or weaponized exploits or arbitrary code execution) are available, then the probability of exploitation for a specific vulnerability is higher than without exploitable features.*

**Assumption 2.** *If a security control is not defined or there is a lack of evidence about the existing control for a specific vulnerability, then the Attack Complexity (AC) can be low and increase the probability of exploitation. Otherwise, AC should be considered based on the CVSS base metrics.*

**Assumption 3.** *If the Access Vector (AV) is a physical or adjacent network and the threat actor has a root or user-level access, then the probability of exploitation can be very high or high.*

**Table 1.** Individual vulnerability assessment.

| Vulnerability Scale | | Description of Vulnerability Level |
| --- | --- | --- |
| **Vulnerability Occurrence** | **Value Range (%)** | **Description of Successful Exploitation of the Vulnerability** |
| Very High (5) | 80–100 | >80% |
| High (4) | 60–80 | 60–80% |
| Medium (3) | 40–60 | 40–60% |
| Low (2) | 20–40 | 20–40% |
| Very low (1) | 1–20 | <20% |

The cumulative chain vulnerability level assessment: This includes the threat actor's exploitation capability to assess a specific vulnerability chain and determines the probability of exploitation for an individual chain. The reason for considering the threat actor's exploitation capability is that the exploitability level of an individual vulnerability may be high, but the threat actor may not have the right capability to exploit the vulnerability due to lack of access vector or attack complexity. Additionally, it is necessary to have knowledge about the specific medical device to exploit a vulnerability related to the medical device. This sub-step measures if a threat actor can successfully reach and exploit each of the vulnerabilities identified in a given vulnerability chain. Figure 1 shows how the threat actor capability is linked to the CVSS metrics and the vulnerability chain. To accomplish this, the calculated individual vulnerability levels and the asset cyber-dependencies produced in the first step and the threat actor profile are considered. Figure 1 also shows that the threat actor capability is linked to the vulnerability chain.

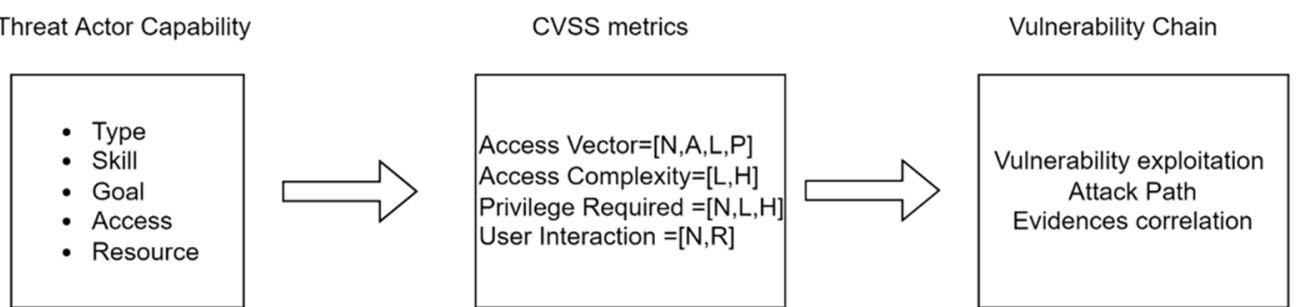

**Figure 1.** Threat actor capability linking with vulnerability chain.

Table 2 shows the threat actor's exploitation capability, which includes the availability of exploitation features and the required access vector for successful exploitation.

**Table 2.** Threat actor exploitation capability.

| Threat Actor Capability Scale | | Description of Scale | | |
|---|---|---|---|---|
| Qualitative Values | Semi-Quantitative Values | Description | Exploitability Features | Metrics |
| Very High | 80–100 | TA has a very sophisticated level of expertise and is well-resourced for the required access vector and attack complexity. TA can generate opportunities to support multiple successful, continuous, and coordinated attacks. | Availability of all features = PoC and Weaponized Exploit, arbitrary code execution | PR = required level Entry point asset AV = required level |
| High | 60–80 | TA has a sophisticated level of expertise, with significant resources for the required access vector and attack complexity. TA has opportunities to support multiple successful coordinated attacks. | Availability of all features = PoC and Weaponized Exploit, arbitrary code execution | PR = required level Entry point asset AV = required level |
| Medium | 40–60 | TA has moderate resources, expertise, and opportunities for the required access vector and attack complexity to support multiple successful attacks. | Availability of some features = PoC and Weaponized Exploit, arbitrary code execution | PR = required level Entry point asset AV = required level |
| Low | 20–40 | TA has limited resources, expertise, and opportunities for the required access vector and attack complexity to support a successful attack. | Availability of some features = PoC and Weaponized Exploit, arbitrary code execution | PR = not required level Entry point asset AV = not required level |
| Very Low | 0–20 | TA has very limited resources, expertise, and opportunities for the required access vector and attack complexity to support a successful attack. | No Availability = PoC and Weaponized Exploit, arbitrary code execution | PR = not required level Entry point asset AV = not required level |

Table 3 presents the cumulative vulnerability level by combining individual vulnerability level and threat actor exploitation capability. The propagated vulnerability assessment estimates how deep into the network an attacker can penetrate if a vulnerability is exploited.

**Table 3.** Cumulative exploitability vulnerability level.

| Threat Actor's Exploitation Capability IVL | Very Low | Low | Medium | High | Very High |
|---|---|---|---|---|---|
| Very Low | VL | VL | L | L | M |
| Low | VL | L | L | M | H |
| Medium | L | L | M | H | H |
| High | L | M | H | H | VH |
| Very High | M | H | H | VH | VH |

**Step 7.3**—Gather and correlate evidence: This sub-step aims to collect relevant evidence that is necessary to consider for the attack path. The approach advocates considering the IoC and related point of compromise (PoC) for the gathering and correlating of the evidence. IoC is a commonly used term for cyber threat intelligence, which broadly indicates unusual behaviour in a system and network. IoCs are the artefacts left due to malicious activity, whereas vulnerabilities are possible weaknesses presented within a system that can be exploited by a threat actor. Evidence of IoC specifies that the vulnerability is already exploited, and the system is compromised. Therefore, the early detection of IoC could delimit the damage of any attack. The possible IoC includes hash code, IP addresses, domains, network traffic, unauthorised setting change, log, suspicious activities on accounts. Additionally, healthcare devices can have other indicators, including configuration changes, disconnection of patient monitors, disruption of healthcare services, or amendment of drug level. Figure 2 shows the possible indicator types for the evidence chain generation.

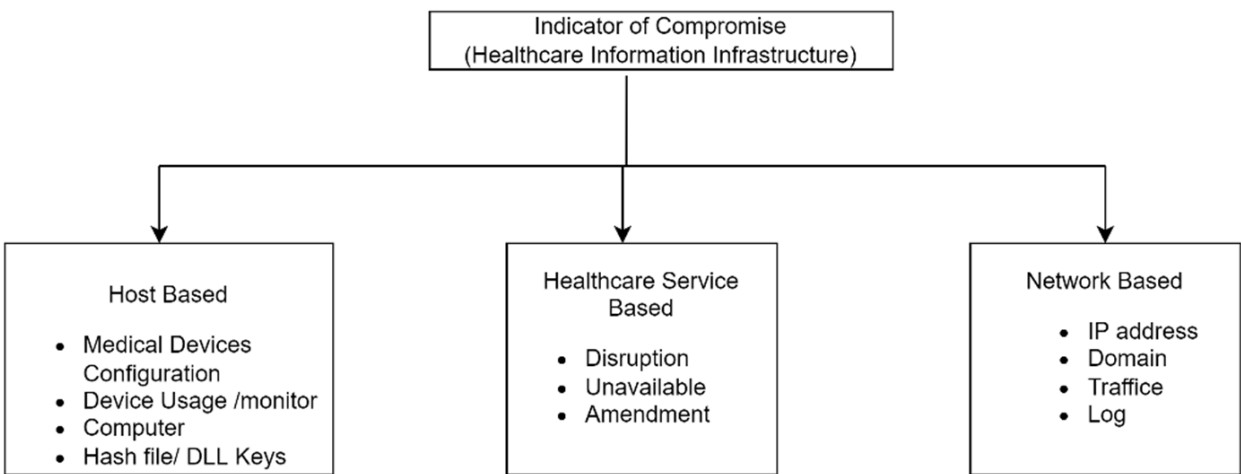

**Figure 2.** Possible IoCs for healthcare information infrastructure.

Once IoC is gathered, then it is necessary to correlate the evidence using the PoC. A PoC is a specific location such as an asset that is compromised by a threat actor. At this stage, it is necessary to determine the common PoC based on the vulnerability and its exploitability within the overall healthcare information infrastructure. The PoC allows us to correlate the IoC to formulate the evidence-based vulnerability chain and reproduce the attack path. The reproducibility also depends on the threat actor's capability to exploit the related vulnerabilities for a successful attack campaign. It is also necessary to determine the level of exploitability for a specific attack path based on the IoC and threat actor's exploitation capability.

**Step 7.4**—Prioritise Attack Path: This is the final sub-step of the proposed method that aims to prioritise the attack paths. The reason for prioritising the attack path is that attack path generation may identify a high number of paths, but some of the paths may not be materialized due to various factors such as lack of exploitability feature, threat actor capability, or a number of security measures in place. Therefore, it is necessary to prioritise the attack paths that are relevant to a specific healthcare context based on the evidence and attacker exploitation capability for the attack path reproduction. The proposed approach exploits the chain level for a confirmed event for this purpose. The prioritisation focuses on the evidence chains which have more chances to the confirmed incident and exhibits potential risks.

Figure 3 shows the attack path generation and analysis process by including the seven defined steps. It considers the overall healthcare ecosystem, which consists of healthcare entities, such as hospitals and clinics, medical and IT devices and healthcare processes and services. This allows attackers to identify the possible entry point and target points for any attack.

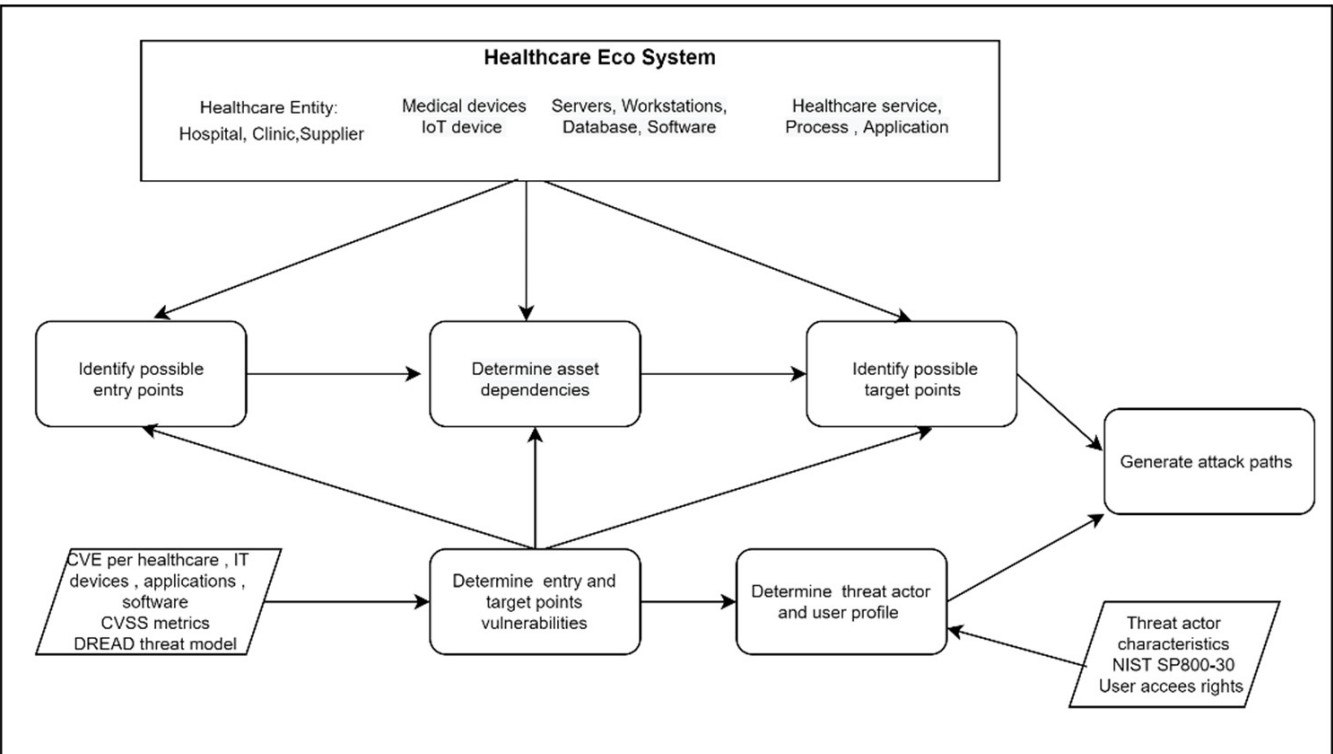

**Figure 3.** Attack path generation and analysis process.

## 4. Attack Path Generation Rules

Rule sets are essential for a successful attack campaign. In particular, the propagation rules are the certain conditions that need to be fulfilled to propagate an attack in different phases. The rules provide certain conditions that are necessary to satisfy and exploit a vulnerability based on the attack profile and asset interdependencies. The rules are created based on different parameters such as attack profile, asset dependency, and vulnerability metrics. The rules are independent of the device specification or the IT infrastructure, so there is no need to amend the rule sets due to the evolution of infrastructure or due to new vulnerabilities. To generate these rules, several variables and the knowledge base (KB) are necessary to be defined and understood.

### 4.1. Variables

The propagation rules determine the possible vulnerabilities that an attacker can exploit for the implementation of a successful attack campaign. Asset dependency, vulnerability exploitability, threat actor profiles and target user profiles are key parameters for propagation rules. A European funded project, named 'A Dynamic and Self-Organized Artificial Swarm Intelligence Solution for Security and Privacy Threats in Healthcare ICT Infrastructures' (AI4HEALTHSEC) [24] works towards developing a solution to enhance the identification and analysis of threats and cyberattacks on healthcare information infrastructures (HCII). The project AI4HEALTHSEC considers a number of attributes for the attack propagation rule set, which are presented below.

**Asset dependency:** An asset within a healthcare ecosystem may have a number of dependencies in order for the related healthcare service to be delivered within the HCII. For instance, an insulin pump needs to inject insulin into the patient's body. The cyber-asset pair enfolds a source cyber-asset and a destination cyber-asset. There are two infrastructures necessary for the transfer and processing of cyber resources, i.e., communications (transmission of big data and information) and IT (use and processing of big data). The dependency type is capable of defining in which manner a cyber-asset pair is interdependent within the healthcare service. To fulfil this, the following cyber dependency types are considered:

exchanging, storing, controlling, processing, accessing, and installing. Additionally, there are also physical dependencies among the assets when one asset is physically connected with another asset.

**Vulnerability exploitability:** Each asset includes a single or multiple vulnerabilities that could be exploited for a given attack path to materialize. Additionally, there are dependencies among the assets that allow us to exploit the vulnerabilities from an entry point asset to a target asset. The attacker needs to identify the entry point asset vulnerabilities that could be discovered and exploited to initiate the attack and then use the target point asset vulnerability to complete the attack path. Therefore, there is a dependency among the assets for the attack path generation.

**Threat actor profile:** The threat actor profile indicates the capability, skills, and motives of an attacker for an attack campaign. Depending on the skill and sophistication, there are variations in the threat actor profile. It is necessary to identify the threat actor profile for the attack path generation. The threat actor profile considers two variables:

- Threat actor capability: Defines the attackers' necessary skill, goals supplication, resources required to execute an attack. It includes five given scales (very low, low, moderate, high, and very high);
- Attack access vector: Define the necessary access path for an attack campaign. There are four different access vectors for executing an attack: **Local**—A vulnerability is exploitable with only local access; **Adjacent**—A vulnerability is exploitable with adjacent network access; **Network**—A vulnerability is exploitable with network access; **Physical**—A vulnerability is exploitable with physical access.

**Target user profile:** Threat actors are targeting various users for an attack campaign. The AI4HEALTHSEC cyberattack path discovery method considers the target user profile that assists the threat actor to execute the attack. The actors involved in the overall healthcare ecosystem, such as healthcare practitioners, nurses, admin workers and IT service workers, could be targeted by a threat actor. The target user profile considers user context such as knowledge and skill (high, medium, and low) for running various healthcare services, applications, and maintenance operations. Users have different execution rights within the system, e.g., an IT admin has the right to install and update applications and a doctor uses different healthcare applications and devices for the healthcare service delivery.

*4.2. Knowledge Base*

As stated before, the attack path identification follows certain rules. However, to generate the rules, it is necessary to define the KB as a foundation for the rule set generation. The KB includes a set of predicate symbols to describe a predicate within the rule set. These will mainly constitute domain elements attributes (e.g., the attributes of assets), along with predicates used in the reasoning process. The list is not exhaustive, and the KB and quantifiers can be extended to capture a more complete or different view of the domain.

Symbols: The following symbols are used for the KB rule set generation:

- Vul denotes Vulnerability which links with an asset;
- Asset denotes specific assets of the overall healthcare ecosystem and possible cyber dependencies with other assets including Hosting, ExchangingData, Storing, Controlling, Processing, Accessing, Installing, Trusted, Inclusion, Interaction, and Connected;
- TA and TAP denotes Threat Actor and Threat Actor Profile, respectively with capability VeryHigh, High, Moderate, Low, and VeryLow, threat require for an attack;
- AV denotes Access Vector with Local Network, Adjacent Network, Local and Physical;
- Vuln_PR denotes Privileged Required as a level of privileges, i.e., None, Low, and High, before successfully exploiting a vulnerability by a Threat Actor;
- Vuln_AC denotes Attack complexity in terms of certain conditions, i.e., Low and High, beyond the attacker's control that must exist in order to exploit the attack;
- Vuln_UI denotes the user interaction, i.e., None and Required, excluding the threat actors for an attack;

- TUP denotes the Target User Profile, i.e., High, Medium, and Low, that assists an attacker to execute an attack;
- Vuln_Exp denotes vulnerability exploitability level, i.e., High, Medium, and Low, based on the exploitability properties;
- Vul_Exp_Fea denotes specific exploitability features of the vulnerability.

The following relationship symbols are used for the KB rule set generation:

Connected relation defines the connectivity between two assets due to the dependency or through the related vulnerabilities. Asset dependencies and inherent vulnerabilities within the asset are considered for the connected relation. Additionally, connectivity can be also achieved if the assets are in the same network. The KB for the connected relation is given below.

Connected using assets dependency.

- $\forall$asset1,asset2 ExchangingData(asset1,asset2) $\lor$ Storing(asset1,asset2) $\lor$ Configuring(asset1,asset2) $\lor$ Updating(asset1,asset2) $\lor$ Accessing(asset1,asset2) $\lor$ Installing (asset1,asset2 $\Rightarrow$ Connected (asset1,asset2) $\land$ Connected(asset2,asset1)

Connected using vulnerability.

- $\forall$vuln1,vuln2,asset1,asset2 Connected(vuln1,asset1, vuln2,asset2) $\Rightarrow$ Connected (vuln2,asset2, vuln1,asset1)

Accessible relation denotes threat actors with specific profiles that can access the asset using a specific access vector required for a confirmed vulnerability.

- $\forall$vuln, asset,TA vuln_AV() $\land$ TAP() $\Rightarrow$ Accessible(vuln,asset,TA)

Exploitable relation denotes threat actors that exploit a specific vulnerability on an asset. The threat actor needs to access the asset for exploitation using the appropriate profile that links with the required base metric values.

- $\forall$vuln,asset,TA Accessible(vuln,asset,TA) $\land$ (vul_UI() $\lor$ vul_PR() $\lor$ vul_AC()) $\land$ TAP() $\Rightarrow$ Exploitable(vuln,asset,TA)

Attacked relation denotes when a threat actor successfully attacks an asset based on a specific vulnerability exploitation and certain profile. Therefore, threat actor accessibility and vulnerability exploitability are required for an attack.

- $\forall$vuln,asset,TA Accessible(vuln,asset,TA) $\land$ Exploitable(vuln,asset,TA) $\Rightarrow$ Attacked (vuln,asset,TA)

*4.3. Attack Path Generation*

4.3.1. Rules Using Access Vector

An existing vulnerability on an asset is accessible by a threat actor based on the possible access vectors such as Network, Adjacent Network, Local, Physical (AV: N/A/L/P).

- If AV is 'Network' (i.e., remotely exploitable), this means both asset and TA are connected to the same network (Internet).
- $\forall$vuln, asset,TA, locNetwork(TA,loc) $\land$ ConnectsTo(asset,loc) $\land$ Vulnerability(vuln,asset) $\land$ Network(vuln) $\Rightarrow$ Accessible(vuln,asset,TA).

Otherwise, if AV is 'Adjacent Network' (i.e., exploitable over local network) and both asset and TA are connected to the same local network.

- $\forall$vuln,asset,TA,loc AdjacentNetwork(TA,loc) $\land$ ConnectsTo(asset,loc).
- $\land$Vulnerability(vuln,asset) $\land$ (AdjacentNetwork(vuln) $\lor$ Network(vuln)) $\Rightarrow$ Accessible(vuln,asset,TA).

4.3.2. Rules Using Base Metrics

The reason for considering vulnerability exploitability is that there are too many confirmed vulnerabilities published each month and it is challenging for healthcare entities to fix all these vulnerabilities. It is necessary to consider the base metrics such as attack vector, attack complexity, privileges required and user interaction for attack path generation. It is

worth mentioning that not all vulnerabilities can be easily exploited due to the nature of the specific product, overall system context and threat actor profile. Additionally, vulnerabilities do not always exploit in isolation, and there is a link between the vulnerabilities and healthcare assets for an attack campaign.

### 4.3.3. Access Vector and Privileges Required

If two vulnerabilities are linked into two different dependent assets, and entry point assets' vulnerability requires AV = N and PR = L and target point assets' vulnerability requires AV = L and PR = N, then TA with AV = N can easily act as a local user to exploit the vulnerability for the target asset. Hence, TA can reach the target asset using the vulnerability of the entry point asset.

Note that if a threat actor obtains (PR = H) for a specific vulnerability on an asset, then TA can exploit the other vulnerabilities on the same asset with lower PR. It implies PR:H $\geq$ PR:L $\geq$ PR:N.

- $\forall$vuln1,asset1,vuln2,asset2, TA(vuln1_AV(N) $\land$ vuln1_PR(L)) $\land$ (vuln2_AV(N) vuln2_PR(L)) $\Rightarrow$ Accessible(vuln2,asset2,TA).

### 4.3.4. Target User Profile and User Interaction

Vulnerabilities often require a certain level of user interaction for successful exploitation. AI4HEALTHSEC correlates the target user profile with the user interaction for this purpose. Generally, three types of user profiles (high, medium, and low) exist depending on knowledge, skill, and experience. If a vulnerability needs user interaction and the target user profile is low for that interaction, this indicates that the user has a lack of knowledge about the context. It is assumed that in such a scenario, the threat actor with a very high and high profile (AC = VH/H) can exploit the vulnerability with the required access vector.

- $\forall$vuln,asset,TA Vuln_UI(R) $\land$ Vuln_TUP(L) $\land$ TA_AC(VH or H) $\Rightarrow$ Exploitable (vuln,asset,TA).

### 4.3.5. Threat Actor Profile and Attack Complexity

If the attack complexity (AC = H) is high, then the threat actor requires a very high or high profile to exploit the vulnerability. For such cases, there are specific conditions beyond threat actor control that are required to be completed before exploitation. A threat actor with very high and high profile is more likely to successfully exploit the vulnerability.

If the threat actor is capable of high AC to trigger an attack on an asset, then it is more likely that the threat actor can exploit also the other low AC on vulnerabilities on the same asset. It implies AC:H $\geq$ AC:L.

- $\forall$vuln,asset, TA Vuln_AC(H) $\land$ Vuln_TAP(VH $\lor$ H) $\Rightarrow$ Exploitable(vuln,asset,TA).

### 4.3.6. Rules Using Vulnerability Exploitability

There are a number of key exploit features, such as proof-of-concept, weaponized, and arbitrary code execution. The exploitability provides the threat actor to reproduce the attack.

### 4.3.7. Exploitability Level and Threat Actor Profile

If the exploitable level for a vulnerability is high, then a threat actor with any profile can attack the specific asset. Additionally, if a low-profile threat actor can successfully attack an asset, then it is more obvious that a threat actor with any other profile level can also attack the asset.

- $\forall$vuln,asset,TA Accessible(vuln,asset,TA) $\land$ Vuln_Exp(H) $\land$ Vuln_TAP(VH $\lor$ H VM $\lor$ L $\lor$ VL) $\Rightarrow$ Attacked(vuln,asset,TA).

### 4.3.8. Proof of Concept Exploit, Weaponized Exploit, Arbitrary Code Execution

There is a strong correlation between the availability of proof of concept and weaponized exploitation for a successful exploitation. Weaponized exploits indicate that the exploit works for every potential threat actor. Additionally, arbitrary code execution also provides more exploitability possibilities.

- $\forall$vuln,asset, TA Vuln_Exp_Fea(PoC ExploitCode) $\wedge$ Vuln_Exp_Fea(weaponized exploits) $\vee$ Vuln_Exp_Fea(arbitrary code execution) $\Rightarrow$ Exploitable(vuln,asset,TA).

## 5. Evaluation: A Healthcare Scenario

The proposed attack path approach is evaluated using a real healthcare case study scenario. This section presents an overview of the scenario, incorporating the implementation of the attack path process. The studied context may identify the potential attacks and take necessary measures to tackle the attacks and related vulnerabilities. The aims of this evaluation are to: demonstrate the applicability of the proposed attack path generation method into a real healthcare scenario; highlight the usefulness of CVSS metrics and exploitability for attack path generation; and display the benefits of the KB rules and IoC for analysing the attack path.

### 5.1. Healthcare Use Case Scenario

The chosen scenario is based on a user-centred Digital Health Living lab, which provides a systematic user co-creation and co-production approach while integrating research and innovation processes in a real-life setting [25]. The residents, council, service providers, academic institutions, and technology companies are the key stakeholders within this living lab and are involved in every step of the way, from the creation of a product or service to commercialization. In particular, the related stakeholders contribute to health innovation in a new way, receive the opportunity to help individuals and society and can be key partners in inspiring health innovation based on their needs, perceptions, and user experience. It is an open innovation ecosystem where the living lab acts as a unique test bed for developing and testing prototypes or more mature digital healthcare solutions. The scenario is mainly based on Tier 3 test and trial category according to the UK National Institute for Health and Care Excellence (NICE) for Digital Health Technologies (DHTs). In particular, Tier 3 aims to help people with a diagnosed condition and provides treatment and health management. It includes tools used for treatment and diagnosis, as well as those influencing clinical management through active monitoring or calculation. This may include a symptom tracking function which records patient information and transmits this to the healthcare team for the derivation and the support of the clinical decision.

Every involved stakeholder, such as patients, healthcare practitioners, residents, and service providers, will engage with the living lab within their own infrastructures and network connections. As such, they connect to the internet through their own Wi-Fi (routers) and communicate through emails (PCs) or their mobile devices (mobile phones, tablets, laptops). There is much critical big data involved in the scenario including patient healthcare information, personal information, device usage and connectivity with other devices. Additionally, the living lab includes various patient healthcare devices such as insulin pump, infusion pump and Internet of Things (IoT) devices for healthcare treatment. The scenario presented above is used to demonstrate the proposed methodology. The next section provides a detailed description of the implementation.

### 5.2. Implementation of the Attack Path Generation

We follow the living lab healthcare scenario to implement the attack path generation (see Figure 4). This section presents the implementation of the attack path generation. Vulnerabilities of healthcare services and systems are the main components for the path generation. In particular, the vulnerabilities in the healthcare sector are unique compared to the other sectors. This is due to the connectivity of different medical devices with the other parts of the network, and these medical devices, in general, have a lack of

security measures. Healthcare information infrastructure contains a large number of legacy systems that are hard to replace and threat actors are always looking into this system for potential exploitation. Healthcare practitioners need to collect sensitive patient data, such as personal and financial information, and therefore potential breaches of this data could provide additional benefits to the criminals or inside attackers.

We made several assumptions for the purpose of implementation. In particular, the home patients use an infusion pump and insulin pump for their treatment and the pump is managed and configured by the healthcare practitioner. Additionally, there are IT devices, such as computers, routers, servers and applications software and operating systems, that are required for the overall system infrastructure. Finally, the low cost of IoT devices, such as smart lamps and IP surveillance cameras, in both home and service provider environments are considered. The security of medical devices is critical to protect patient information and to ensure healthcare service delivery since the devices are connected to the internet. These devices are dependent on the other IT devices and network infrastructure to exchange and collate data from various sources for making clinical decisions. There are vulnerabilities due to the interdependencies among the assets from the hardware, software, human, and overall healthcare system context. Compromised healthcare devices can be used to propagate the attack path on the other part of the healthcare information infrastructure. Software is embedded in the devices to assist their functions and operation of the medical devices. Therefore, an attack path can also be initiated and propagated from this embedded software. Additionally, web services are commonly used for interfacing the connected medical devices with the other parts of the system.

A list of assets is identified based on the scenario which consists of medical devices, IT devices, IT infrastructure and applications. These assets are critical for the overall healthcare service delivery and research activities for the living lab. The potential threat actors and user profiles are also considered to demonstrate the attack path. We have extracted a number of vulnerabilities from the CVE database and categorised them based on the assets of the studied scenario. Additionally, CVSS is also considered for the base metrics properties which are necessary for the ruleset. The identified vulnerabilities and base metrics are used to generate the attack paths. The process allows the generation of a possible attack path and the CVSS metrics impact value is considered to select the appropriate ones. The attack path generation is iterative to generate the possible attack paths.

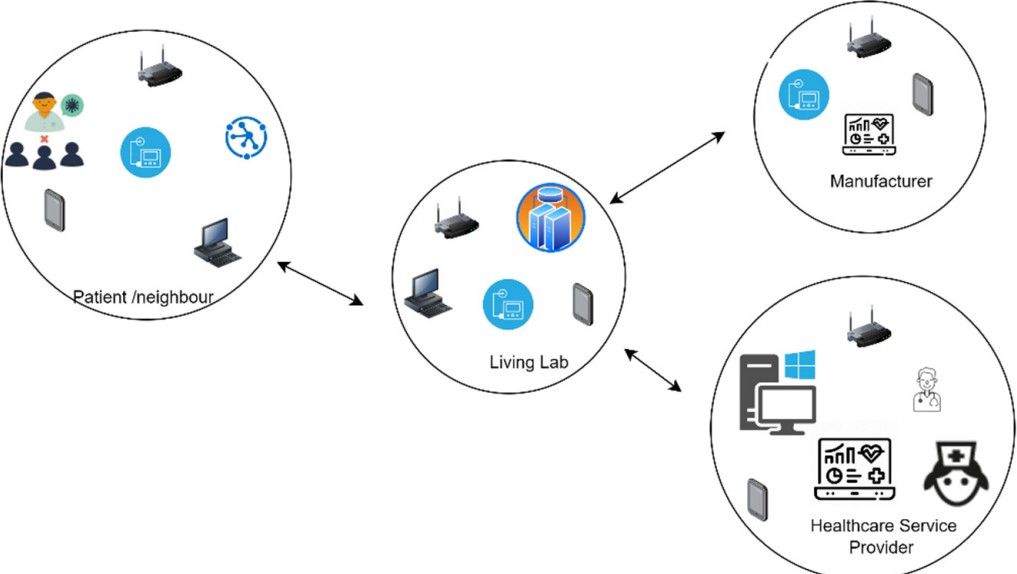

**Figure 4.** Living lab healthcare scenario.

### 5.2.1. Identify Entry, Target Point Assets, and Possible Dependencies

This section combines the first three steps of the attack paths, providing a list of assets identified based on the healthcare use case scenario, specifically, the healthcare service delivery and related healthcare information infrastructure. Each device needs an interface such as a wireless or network interface to connect with the other devices. For instance, an insulin pump management system that is physically located at home has one wireless interface to interact with another device interface. Note that we have only considered the devices for the demonstration of the attack path. Entry and Target Point Assets: Medical and IT device:

- **Infusion Pump (A1):** Braun's Infusion System 871305U aims to deliver fluid such as nutrients and medications into the patient's body. Trained healthcare practitioners should program the rate and duration for the medication. The pump stores patient drug information;
- **Insulin pump (A2):** Medtronic MiniMed 508 is one of the most widely used pumps for delivering a specific amount of insulin to the diabetic patient's body. The device is programmed to inject a specific amount of insulin set by the doctor into the patient body. The pump stores patient sensitive insulin information;
- **Insulin management system (A3):** Omnipod DASH Insulin Management System 19191 is a tubeless and wireless system that allows continuous insulin delivery for 3 days. It consists of a pod that is worn directly on the patient's body and a personal diabetic manager which programs and controls the delivery;
- **IoT devices (A4):** There are several IoT devices that are relevant to the scenario. A heartrate monitor (Maxim's 700-MAXREFDES117) can be used to monitor the heart rate (wearable device). Additionally, a smart light system (Philips) is also considered for the healthcare service delivery;
- **Information and communication network (A5):** This includes multiple devices, such as routers, WiFi, switches, wireless interface cards, and others that are responsible for the connection from the device to the network;
- **Computer system (A6):** Windows-based workstation and servers connected to the medical devices, patient interfaces and servers;
- **Rugged tablets (A7):** These tablets are commonly used for patient care applications such as medication alerts and tracking, Electronic Health Records (EHR) support, blood pressure monitoring, and connecting to barcode readers and can directly interface to the other medical equipment. Healthcare practitioners can directly use these tablets for patient treatment.

### 5.2.2. Information and Software

- **Hospital information management system (A8):** Care 2X software is a patient medical record and staff management system. It supports web-based platforms and a simple user interface. The patient medical record includes patients' identifiable and treatment information;
- **SpaceCom and SpaceStation (A9):** This is the software that operates the infusion pump and resides either on the pump or the space station. Generally, the pump is attached to the space station. We have considered Braun's Infusion System 871305U, which is linked with the SpaceCom 012U000050. SpaceCom is responsible to update two critical functions, i.e., drug library and pump configuration. Drug libraries can prevent incorrect dosing of drugs;
- **Device usage information:** This includes the amount of time used by the patient from the device and the relevant programme data.

### 5.2.3. Possible Asset Dependencies

The identified assets rarely perform any operation alone. Assets within the healthcare system are connected for a specific service delivery. For instance, the data from the home infusion pump are transferred to the pump server. The server correlates the data for making

clinical decisions. The home care service software needs to update the medical device installed into the home healthcare system. The insulin pump needs to inject insulin into the patient's body and is controlled by the software through wireless communication. Therefore, there are different types of dependencies among the assets, which are shown in Table 4.

**Table 4.** Asset dependency.

| Entry Point Asset and Type | Target Point Asset and Type | Dependency Type |
| --- | --- | --- |
| A9 (SpaceCom Software) | A1 (Infusion Pump) | Configured_to, Updated_to |
| A3 (Insulin Management System) | A2 (Insulin Pump) | Configured_to, Updated_to |
| A8 (Care2X-Hospital Management System) | A6 (Windows System) | Installed_on, Updated_by |
| A5 (Router) | A6 (Windows System) | Connected_to, Exchange_data |
| A4 (IoT Device) | A5 (Router) | Connected_to, Exchange_data |
| A7 (Tablet) | A8 (Care2X-Hospital Management System) | Exchange_data |
| A9 (SpaceCom Software) | A1 (Infusion Pump) | Configured_to, Updated_to |

The asset dependency graph is also presented in Figure 5.

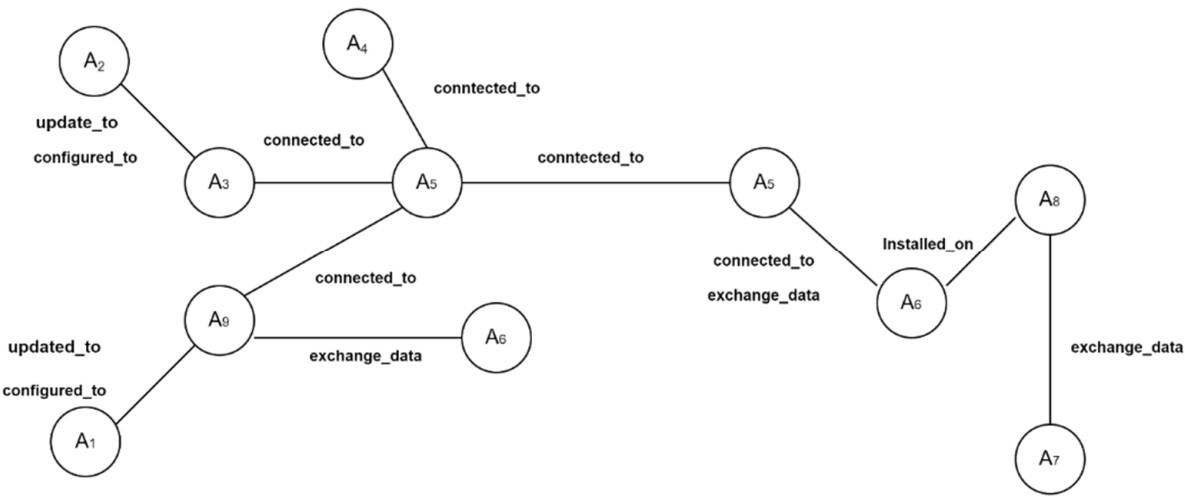

**Figure 5.** Asset-dependency graph.

5.2.4. Entry and Target Point Vulnerabilities

Once the assets and their dependencies are identified, the next step is to identify the vulnerabilities that can be exploited in order to compromise the assets. As mentioned before, the CVE list is used for vulnerability identification. There are fifteen confirmed recent vulnerabilities considered among those assets which go as follows:

- two vulnerabilities are identified on asset A1;
- one vulnerability is identified on asset A2;
- one vulnerability is identified on asset A3;
- one vulnerability is identified on asset A4;
- three vulnerabilities are identified on asset A5;
- three vulnerabilities are identified on asset A6;
- one vulnerability is identified on asset A7;
- one vulnerability is identified on asset A8;
- two vulnerabilities are identified on asset A9.

Once the vulnerabilities are identified, it is necessary to understand the base nature of exploitability and the base metric for each specific vulnerability. This allows for the analysis regarding how the asset of this scenario can be exploited considering the threat actor profile. Table 5 presents details regarding the identified vulnerabilities.

**Table 5.** Vulnerabilities and CVSS metrics for each asset.

| Asset | Vulnerabilities & Exploitability |
|---|---|
| A1 = Braun's Infusion Pump | A1,V1 = Lack of input validation provides command line access and privilege escalation. TA requires in the same network as device<br>CVE-2021-33886, A1.V3 = VH(8.8)<br>AV:A/AC:L/PR:N/UI:N/S:U/C:H/I:H/A:H |
| | A1,V2 = Unrestricted file upload that can overwritten critical files due to privilege escalation<br>CVE-2021-33884, A1.V4 = VH(9.1)<br>CVSS:3.1/AV:N/AC:L/PR:N/UI:N/S:U/C:N/I:H/A:H |
| A2 = Medtronic MiniMed 508 insulin pump | A2,V1 = lack of security (authentication and authorization) in RF communication protocol with other devices such as blood glucose meter and glucose sensor transmitters. TA requires in the same network as device can inject or intercept data and change pump settings<br>CVE-2019-10964, A2.V1 = VH(8.8)<br>CVSS:3.0/AV:A/AC:L/PR:N/UI:N/S:U/C:H/I:H/A:H |
| A3 = Insulin Management System | A3,V1 = improper access control in the wireless RF communication protocol allows local TA to intercept or modify insulin data and change pump settings.<br>CVE-2020-10627 A3.V1 = VH(8.1)<br>CVSS:3.1/AV:A/AC:L/PR:N/UI:N/S:U/C:H/I:H/A:N |
| A4 = IoT device Philips Hue light bulb | A4,V1 = communication protocol can be abused to remotely installed malicious firmware in the light bulb as remote code execution through buffer overflow and spread to other IoT devices that use Zigbee communication protocol.<br>CVE-2020-6007 A4.V1 = H(7.8)<br>CVSS:3.1/AV:A/AC:H/PR:N/UI:R/S:C/C:H/I:H/A:H |
| A5 = Router (Buffalo, Cisco RV Series—Netgear) | A5,V1 (Buffalo routers) = Bypass authentication procedures on the affected routers though files which do not need authentication and gain root level access. It enables telnet service to connect other devices' control such as IoT Devices.<br>CVE-2021-20090 A5.V1 = VH(9.8)<br>CVSS:3.1/AV:N/AC:L/PR:N/UI:N/S:U/C:H/I:H/A:H |
| | A5,V2 (Cisco RV series) = Remote TA with administrative privileges inject arbitrary commands into operating system due to lack of input level validation through web-based interface.<br>CVE-2021-4012 A5.V2 = H(7.2)<br>CVSS:3.1/AV:N/AC:L/PR:H/UI:N/S:U/C:H/I:H/A:H |
| | A5,V3 (Netgear router) = unauthenticated TA can affect the device through buffer overflow attack.<br>CVE-2018-21224 A5.V3 = VH(8.8)<br>CVSS:3.1/AV:A/AC:L/PR:N/UI:N/S:U/C:H/I:H/A:H |
| A6 = System (Windows Compatible) | A6,V1 = A remote code execution vulnerability allows TA to execute arbitrary code and gain same right as current user. This allows to install program modify files based on the existing user rights.<br>CVE-2019-1236 A6.V1 = H(7.5)<br>CVSS:3.1/AV:N/AC:H/PR:N/UI:R/S:U/C:H/I:H/A:H |
| | A6,V2 = A remote code execution vulnerability allows TA to execute arbitrary code and gain same right as current user. TA needs control of a server to execute this vulnerability and tricks the user for the to connect the server.<br>CVE-2019-1333 A6.V2 = H(7.5)<br>CVSS:3.1/AV:N/AC:L/PR:N/UI:R/S:U/C:H/I:H/A:H |
| | A6,V3 = A remote code execution vulnerability allows TA to run arbitrary code with system privilege. TA could install program, amend files, and create new users with full rights.<br>CVE-2021-36958 A6.V3 = H(7.8)<br>CVSS:3.1/AV:L/AC:L/PR:N/UI:R/S:U/C:H/I:H/A:H |
| A7 = Rugged Tablet (Dell) | A7,V1 = A local TA without the necessity of authentication can exploit this vulnerability and execute arbitrary code in system management mode.<br>CVE-2020-5348 A8.V1 = H(7.8)<br>CVSS:3.1/AV:L/AC:L/PR:L/UI:N/S:U/C:H/I:H/A:H |
| A8 = web-based hospital management system (Care2X) | A8,V1 = A cross site scripting vulnerability exploited during patient registration. TA can send the XSS payload to this vulnerable parameter and take control of another register user. TA needs victim user interaction<br>Exploitability features = PoC and Weaponized Exploit,<br>CVE-2021-36352 A9.V1 = M(5.4)<br>CVSS:3.1/AV:N/AC:L/PR:L/UI:R/S:C/C:L/I:L/A:N |
| A9 = SpaceCom | A9,V1 = Lack of authentication for critical space com function allows connection to the pump<br>Exploitability features = PoC and Weaponized Exploit, arbitrary code execution<br>CVE-2021-33882, A1.V1 = VH(8.6)<br>AV:N/AC:L/PR:N/UI:N/S:C/C:N/I:H/A:N |
| | A9,V2 = Clear text transmission allows TA to snoop network traffic.<br>Exploitability features = PoC and Weaponized Exploit, arbitrary code execution<br>CVE-2021-33883, A1.V2 = H(7.5)<br>AV:N/AC:L/PR:N/UI:N/S:U/C:H/I:N/A:N |

### 5.2.5. Threat Actor and User Profile

The threat actor profile considers capability and attack vectors for exploiting a vulnerability. The threat actors can be external or internal with different motivations such as financial gain, harm to the patient, and/or competitors. In general, a threat actor needs to understand the device-specific verification information, and specifically for the medical devices, it is necessary to understand spectrum, transmission radio frequency, and data structure.

- **Infusion Pump:** TA should have knowledge regarding the access to the local network, CAN bus data structure, escalation of privilege from user access to admin access and the pump configuration;
- **Medtronic Insulin Pumps:** Knowledge regarding how to access the network and intercept radio frequency and the pump configuration;
- **Smart bulb:** Knowledge of smart bulb operation and access point to overtake the bulb control.

Finally, healthcare practitioners and other users need to perform various activities based on the roles for the healthcare service delivery. For instance, a practitioner needs to update the patient's medical records, set insulin levels, and monitor infusion pump activities for the service delivery. Therefore, the practitioner needs to have basic knowledge about how to operate the devices and their security. IT users need to update and manage all devices, including medical and IT, within the network.

*5.3. Results: Attack Path Generation*

Once the asset dependencies and vulnerabilities are identified, this final step aims to generate the attack path. There are a number of attack paths generated from the source asset to the target asset through the combination of vulnerabilities and dependencies. Note that there can be additional attack paths generated from the scenario, but this section presents only the relevant ones. Additionally, we only considered the three critical target point assets, i.e., infusion pump, insulin pump, and healthcare system, for the attack path generation.

Attack Path—Target Point Infusion Pump: It is assumed that the threat actor (TA) is acting as an outsider without any prerequisite credential (PR:N) and user interaction (UI:N) can gain user level access to the SpaceCom system (A9,V1) through the network (AV:N) and escalate the privileges to gain root access. This allows the TA to communicate with the pump (A1,V1) with no privilege (AV:A) and user interaction (UI:N). The TA can finally manipulate the drug library or pump configuration. The TA can also execute malicious code in the pump's RTOS by accessing the SpaceCom (A9,V2) and executing the code and overwrite the pump (A1,V2) RTOS. Additionally, a TA who is able to access the hospital management system can obtain the patient drug information and further exploit the infusion pump. The TA can also exploit the hue light bulb (A4,V1) to access the home network and then further propagate into the infusion pump. This can happen mainly when the pump is idle or in standby mode. There are four potential attack paths through which the target point infusion pump can be exploited:

A5,V1 → A9,V1 → A1,V1
A9,V2 → A1,V2
A8,V1 → A9,V1 → A1,V2
A4,V1 → A9,V1 → A1,V1

Attack Path—Target Point Insulin Pump: It is assumed that an internal threat actor (i.e., may be an employee) who has access to the insulin management system (A3,V1) using (AV:A/L), unrestricted (user) access (PR:N), basic computer skills (AC:L) and without user interaction (UI:N) exploits the insulin pump (A2,V1). The TA can also exploit the router (A5,V3) through an adjacent network and access the insulin management system (A3,V1) to access the pump (A2,V1). A TA from an adjacent network can take control of the hue lightbulb (A4,V1), become unreachable to the user and send malicious code to other devices and networks. It is assumed that a patient as the user may have a lack of knowledge about the smart bulb operation, which can be exploited by the TA. When the user interacts with the bulb, the TA can take over the control and propagate to the other part of the network. There are three potential attack paths which can exploit the target point insulin pump.

A5,V3 → A3,V1 → A2,V1
A3,V1 → A2,V1
A4,V1 → A3,V1 → A2,V1

Attack Path—Target Point Hospital Management System: It is assumed that a TA with network access through the router (A5,V2) can exploit the healthcare system (A8,V1) that is installed on a Windows-based system (A6,V1). User interaction is necessary to exploit this attack path; therefore, a healthcare practitioner needs to interact with the system for the exploitation. Additionally, such an attack path needs a TA with high skills who needs root level privilege to exploit the router and amend the user rights within the windows system. It allows them to access the hospital management system and add new users and gather sensitive data from the system. Another possibility could be that an internal TA with local access from tablet (A7,V1) may also attempt to exploit the (A9,V1) through the windows system (A6,V3). This path needs a local access vector and user interaction. There are three potential attacks through which the target point infusion pump can be exploited.

A5,V3 → A6,V1 → A8,V1
A5,V3 → A6,V2 → A8,V1
A7,V1 → A6,V3 → A8,V1

### 5.4. Generate and Prioritise Evidence-Based Vulnerability Chain

We assume that there are two confirmed events that occurred in the studied living lab scenario. The first event occurred in the patient homecare unit, where cyber threats are detected on the home care and the IoCs are analysed. In the current scenario, unauthorized access to the SpaceCom software (PoC = A9) allows the threat actor to access the infusion pump (PoC = A1). The IoCs in this case are the log (IoC1), amendment of drug library (IoC2), pump configuration (IoC3), and obtained pump data (IoC4). This case enfolds a confirmed event of a cyberattack. To discover and produce the potential cyberattack paths for the compromised asset A1, cyber dependency with the infusion pump (A1) is considered, and possible attack paths for A1 are listed. The second event is data leak, where high-profile TAs access the hospital management system (PoC = A8) and collect the data (IoC6) through Windows system (PoC = A6) using IoC5 and IoC7 (user right and install program). Cyber threats are detected in the healthcare service provider infrastructure. Table 6 shows the evidence chain for the identified security incidents.

**Table 6.** Attack path based on confirmed security events and potential evidence chains.

| Security Incident | Attack Path | Evidence Chain |
|---|---|---|
| Amendment of drug level and pump configuration | A5,V1 → A9,V1 → A1,V1<br>A9,V2 → A1,V2<br>A8,V1 → A9,V1 → A1,V2<br>A4,V1 → A9,V1 → A1,V1 | A5,V1 → A9, IoC1 → A1, IoC2<br>A9, IoC4 → A1, IoC3<br>A8,V1 → A9, IoC1 → A1, IoC2<br>A4,V1 → A9, IoC1 → A1, IoC2 |
| Patient data leak | A5,V3 → A6,V1 → A8,V1<br>A5,V3 → A6,V2 → A8,V1<br>A7,V1 → A6,V3 → A8,V1 | A5,V3 → A6, IoC5 → A8, IoC6<br>A5,V3 → A6, IoC7 → A8, IoC6<br>A7,V1 → A6,V3 → A8, IoC6 |

To estimate the exploitability for each reconstructed attack path, different attackers' profiles are considered and displayed in Table 7. To estimate the EL per vulnerability, with respect to the analysed IoC, only vulnerabilities of the assets interconnected with the IoCs are considered.

Once the individual vulnerability and TA exploitability level are identified, then it is necessary to determine the probability of an attack path exploitation level. This needs to consider IoCs related to the attack path. Note that the probability of exploitation for a disclosed IoC is the maximum value; therefore, exploitation level for the attack path depends on the vulnerabilities that are not exploited. These values are converted to qualitative values to estimate the attack path exploitability level (APEL) defined in the previous section. This is depicted in Table 8 for APEL. We made a number of assumptions for a given vulnerability based on the CVSS metrics. For instance, attack path 1 in the initial node, A5,V1, and threat actor capability is considered as a medium to exploit the

vulnerability. Therefore, the exploitation level for the A5,V1 is H by following Table 8 and APEL for the overall attack path is H. Another example could be attack path 2, where both nodes are exploited; therefore, the APEL should be the maximum value. The TA capability for the A4,V1 is low, A5,V3 is medium A7,V1 is high, and A6,V3 is medium.

**Table 7.** Individual vulnerability exploitation.

| Vulnerability | Asset | Individual Vulnerability Level (IVL) | Threat Actor's Exploitability Level | | | | |
| --- | --- | --- | --- | --- | --- | --- | --- |
| | | | Capability = Very Low (VL) | Capability = Low (L) | Capability = Moderate (M) | Capability = High (H) | Capability = Very High (VH) |
| V1 | A1 | IVL(A1,V1) = VH | M | H | H | VH | VH |
| V2 | A1 | IVL(A1,V2) = VH | M | H | H | VH | VH |
| V1 | A9 | IVL(A9,V1) = VH | M | H | H | VH | VH |
| V2 | A2 | IVL(A9,A2) = H | L | M | H | H | VH |
| V1 | A6 | IVL(A6,V1) = H | L | L | H | H | VH |
| V2 | A6 | IVL(A6,V2) = H | L | L | H | H | VH |
| V1 | A8 | IVL(A8,V1) = VH | M | H | H | VH | VH |

**Table 8.** Prioritised attack path.

| AP No. | Attack Paths | Evidence Chains | Exploitation Level Chain (ELC) | Exploitation Probability | APEL |
| --- | --- | --- | --- | --- | --- |
| 1 | A5,V1 → A9,V1 → A1,V1 | V1 → IoC1 → IoC2 | H → IoC1 → IoC2 | 0.75 × 1 × 1 = 0.75 | H |
| 2 | A9,V2 → A1,V2 | IoC4 → IoC3 | IoC4 → IoC3 | 1 × 1 = 1 | VH |
| 3 | A8,V1 → A9,V1 → A1,V2 | V1 → IoC1 → IoC2 | M → IoC1 → IoC2 | 0.5 × 1 × 1 = 0.5 | M |
| 4 | A4,V1 → A9,V1 → A1,V1 | A4,V1 → A9, IoC1 → A1, IoC2 | M → IoC1 → IoC2 | 0.5 × 1 × 1 = 0.5 | M |
| 5 | A5,V3 → A6,V1 → A8,V1 | A5,V3 → IoC5 → IoC6 | H → IoC5 → IoC6 | 0.75 × 1 × 1 = 0.75 | H |
| 6 | A5,V3 → A6,V2 → A8,V1 | A5,V3 → A6, IoC7 → A8, IoC6 | H → IoC7 → IoC6 | 0.75 × 1 × 1 = 0.75 | H |
| 7 | A7,V1 → A6,V3 → A8,V1 | A7,V1 → A6,V3 → A8, IoC6 | H → H → IoC6 | 0.75 × 0.75 × 1 = 0.56 | M |

## 6. Discussion

The purpose of this research was to present the attack path discovery method considering the unique characteristics of the healthcare information infrastructure, such as assets and their cyber and physical dependencies, vulnerabilities, threat actor and user profile, and IoC. A scenario in a real-life healthcare setting has also been used to prove the implementation of the attack path discovery method. The cyber threat landscape is constantly evolving, and threat actors are highly skilled in conducting sophisticated and multiple attacks on a number of infrastructures. They target the initial access point assets and exploit possible vulnerabilities to reach the target point through several intermediate nodes.

Research shows that most studies on cybersecurity in the healthcare field focus on technical aspects [26]. Following this focus on technology, other significant components, such as threat actors' profiles and related psychosocial and behavioural characteristics remain understudied in the field [20]. This comes as a surprise when taking into consideration that most cyberattacks are caused by individuals and the adopted risk mitigation by technological solutions is successful to a limited extent. The core of a sturdy strategy for cyberattacks needs to be human-centric and consider attackers' profiles for ultimate benefit. It is worth noting that attack potentials are positively connected to attackers' profiles, while studying this further would shed light on the early detection, prevention, and protection of cybersecurity incidents within healthcare organizations. Examining involved human aspects is also of paramount importance to further investigate how healthcare professionals understand data privacy and security and its significance. This significance lies on the attitudes towards cyber threats, operations, and related controls.

The proposed methodology provides an understanding of possible entry point assets for the studied context and possible paths to reach the target point asset. It adopts the CVSS metrics and its exploitability feature for a common understanding of how the threat actor can exploit particular attack paths. Additionally, threat actor individual and exploitability capability are also taken into consideration for attack path generation and prioritisation.

Hence, the combination of threat actor capability, i.e., skill, motivation, and location, with the availability of exploitability features justifies the prioritised attack paths. Healthcare information infrastructure is an attractive target for the threat actor due to the potential benefits of obtaining sensitive patient data. In recent years, the value of personal medical data has increased on the black market. Credit card information sells for USD 1–2 on the black market, but personal health information (PHI) can sell for as much as USD 363. Therefore, the proposed attack path discovery and prioritisation provides an effective way to identify the potential attacks and undertake suitable control to tackle the attacks.

Cybersecurity issues should be considered from the design stage of the medical devices, otherwise risks will continue to grow. Medical devices are no longer a standalone component, but they are rather connected with other devices for overall healthcare service delivery. Vulnerabilities in connected devices used in hospital networks would allow attackers to disrupt healthcare service delivery and medical equipment. There is a need for sound and proven cybersecurity approaches for ensuring overall security. Threat actors tend to exploit vulnerabilities within a network and form attack paths from one asset to another until they have reached the asset they wish to harm. The proposed approach assists in identifying the common vulnerabilities that can be exploited within the healthcare context so that the necessary course of action can be taken into consideration.

## 7. Conclusions

Enhancing the security and resilience of healthcare service delivery is of paramount importance for securing the overall healthcare ecosystem. It is always necessary to ensure the safety of patients' data and secure healthcare service delivery. The proposed approach provides an understanding of the areas that have potential for cyberattacks. This is conducted by looking for existing vulnerabilities and their possible exploitations based on the assets and their dependencies for possible attack path generation. This work contributes to the identification of the vulnerabilities from both healthcare and IT devices and demonstrates how the attack paths can be propagated from a connected medical device to other parts of the system. This can also be possibly achieved in other infrastructures and scenarios, identifying the relevant attack areas and deploying appropriate measures. The novelty of the proposed approach is to analyse the threat actor profile to generate attack paths and use evidence-based vulnerability chain to prioritise the attack path. This allows us to determine the suitable control to tackle the attacks. Finally, the approach is applied to the living lab healthcare scenario, and the results from the studied context identify the possible attack paths based on the asset and related vulnerabilities. These paths are prioritised so that suitable controls can be identified to tackle the attack for secure healthcare service delivery. As part of our future research, we would like to deploy the proposed methodology in different healthcare context and other supply chain system. Additionally, it is necessary to develop a checklist of controls that would link with the attack paths for the overall cyber security improvement.

**Author Contributions:** Conceptualization, S.I., S.P., E.-M.K. and K.K.; methodology, S.I. and S.P.; validation, S.I. and K.K.; formal analysis, S.I., S.P. and E.-M.K.; investigation, S.I., S.P., E.-M.K. and K.K.; resources, S.I., S.P., E.-M.K. and K.K.; writing—original draft preparation, S.I.; writing—review and editing, S.I. and K.K.; visualization, S.I., S.P., E.-M.K. and K.K.; supervision, S.I.; project administration, K.K.; funding acquisition, S.P. All authors have read and agreed to the published version of the manuscript.

**Funding:** The research conducted in this paper was triggered by the authors' involvement in the project 'A Dynamic and Self-Organized Artificial Swarm Intelligence Solution for Security and Privacy Threats in Healthcare ICT Infrastructures' (AI4HEALTHSEC) under grant agreement No 883273. The authors are grateful for the financial support of this project that has received funding from the European Union's Horizon 2020 research and innovation programme. The views expressed in this paper represent only the views of the authors and not of the European Commission or the partners in the above-mentioned project.

**Conflicts of Interest:** The authors declare no conflict of interest. The funders had no role in the design of the study; in the collection, analyses, or interpretation of data; in the writing of the manuscript, or in the decision to publish the results.

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
