# Peer review of "Cyberattack Path Generation and Prioritisation for Securing Healthcare Systems"

_applsci, doi:10.3390/app12094443_

Round 1

Reviewer 1 Report

 Authors try a method to identify and analyze the possible attack paths to secure the healthcare ecosystem, based on potential vulnerabilities in the system.

Weaknesses/issues to be addressed:

  1. Line 266 on page 6: ” This step of the described methodology aims to generate the possible attack paths. The aim is to gather attack paths against target point assets...” Sentences are repetitive and need to be concise. Similarly, Line 54 and Lin 60 on Page 2, “Indicator of Compromise (IoC)” repetitive appear “IoC” abbreviation.

  1. Tables (or Figures) are not written uniformly, for example, Line 307 and line 327 on Page 7, ” Table: ???” and “Table.???”. Similarly, Line 351 on Page 9: “Figure 2: Possible IoCs for...” and “Figure 2. ???”

  1. Subtitle numbers are messed up, such as: Line 886 on Page 25:” 4. Discussion” should be6. Discussion”, and ” 5. Conclusions” should be ” 7. Conclusions”. I guess Line 846 on Page 9 “5.3.1” should be “5.4”
  2. The same sentence appear in different paragraphs, such as Line 161 on Page 4 ”A network attack path approach based on vulnerability correlation, analyzing…”is same as that Line 75 on Page 2. It should be concise and clear in paper wording.

Author Response

Authors’ response to Reviewers’ Comments

Applied science

MDPI

Manuscript ID: applsci-1635858

Title: Cyber Attack Path Generation and Prioritisation for Securing the Healthcare Systems

We want to thank the reviewers for their time and effort to review our paper and for their comments. The new version of the paper addresses all the comments raised by the reviewers. The text below indicates our responses to those comments and points out areas of the paper that have been modified based on the comments. We have added both track change and without track change version of the revised paper.

To facilitate a more straightforward reading of our responses, we have used the following format in the text below:

  • Each comment has been highlighted in bold.
  • After each reviewer’s comment, our corresponding reply is provided.

Reviewer 1 comments

Review Report Form

Open Review

English language and style

( ) Extensive editing of English language and style required
( ) Moderate English changes required
( ) English language and style are fine/minor spell check required
(x) I don't feel qualified to judge about the English language and style

Yes

Can be improved

Must be improved

Not applicable

Does the introduction provide sufficient background and include all relevant references?

( )

( )

(x)

( )

Is the research design appropriate?

( )

( )

(x)

( )

Are the methods adequately described?

( )

( )

(x)

( )

Are the results clearly presented?

( )

(x)

( )

( )

Are the conclusions supported by the results?

( )

( )

(x)

( )

Comments and Suggestions for Authors

Comment :  Authors try a method to identify and analyze the possible attack paths to secure the healthcare ecosystem, based on potential vulnerabilities in the system.

Weaknesses/issues to be addressed:

Comment:   Line 266 on page 6: ” This step of the described methodology aims to generate the possible attack paths. The aim is to gather attack paths against target point assets...” Sentences are repetitive and need to be concise. Similarly, Line 54 and Lin 60 on Page 2, “Indicator of Compromise (IoC)” repetitive appear “IoC” abbreviation.

 Response:   We thank the reviewer for their feedback and comments. This has now been removed to ensure conciseness of the text.

Comment : Tables (or Figures) are not written uniformly, for example, Line 307 and line 327 on Page 7, ” Table: ???” and “Table.???”. Similarly, Line 351 on Page 9: “Figure 2: Possible IoCs for...” and “Figure 2. ???”

Response:  All captions have now been corrected according to the journal’s guidelines.  

Comment : Subtitle numbers are messed up, such as: Line 886 on Page 25:” 4. Discussion” should be”6. Discussion”, and ” 5. Conclusions” should be ” 7. Conclusions”. I guess Line 846 on Page 9 “5.3.1” should be “5.4”

Response: This has now been corrected.  

Comment: The same sentence appear in different paragraphs, such as Line 161 on Page 4 ”A network attack path approach based on vulnerability correlation, analyzing…”is same as that Line 75 on Page 2. It should be concise and clear in paper wording.

 Response: This has been removed.

Reviewer 2 Report

This paper is presented in a very interesting subject. Indeed the focus of the paper is adequate and a lot of work must be performed in this subject.

However, before being published if accepted, authors must perform changes solving some issues, namely:

  • Paper is too descriptive! Authors should reduce unuseful descriptions. Because of this, soem readers loose interest for this interesting subject;
  • Paper is too analytic; some more real applications must be presented for validation of the presented approach for "Cyber Attack Path Generation and Prioritisation for Securing the Healthcare Systems";
  • The main added value of this work - when compared with other similar ones - is not clearly presnted to the readers! Authors must be more objective in their sentences and explanations;
  • Conclusions must be improved and more objective about the main added value of the paper.

If some real improvements will be performed, perhaps it could be an interesting paper, due to the subject, very important nowadays. 

Author Response

Authors’ response to Reviewers’ Comments

Applied science

MDPI

Manuscript ID: applsci-1635858

Title: Cyber Attack Path Generation and Prioritisation for Securing the Healthcare Systems

We want to thank the reviewers for their time and effort to review our paper and for their comments. The new version of the paper addresses all the comments raised by the reviewers. The text below indicates our responses to those comments and points out areas of the paper that have been modified based on the comments. We have added both track change and without track change version of the revised paper.

To facilitate a more straightforward reading of our responses, we have used the following format in the text below:

  • Each comment has been highlighted in bold.
  • After each reviewer’s comment, our corresponding reply is provided.

Open Review 2

English language and style

( ) Extensive editing of English language and style required
( ) Moderate English changes required
(x) English language and style are fine/minor spell check required
( ) I don't feel qualified to judge about the English language and style

Yes

Can be improved

Must be improved

Not applicable

Does the introduction provide sufficient background and include all relevant references?

( )

( )

(x)

( )

Is the research design appropriate?

( )

(x)

( )

( )

Are the methods adequately described?

( )

( )

(x)

( )

Are the results clearly presented?

( )

( )

(x)

( )

Are the conclusions supported by the results?

( )

( )

(x)

( )

Comments and Suggestions for Authors

Comment: This paper is presented in a very interesting subject. Indeed the focus of the paper is adequate and a lot of work must be performed in this subject. However, before being published if accepted, authors must perform changes solving some issues, namely:

Response:   We thank the reviewer for their feedback and comments.  All the raised issues are addressed accordingly. 

Comment: Paper is too descriptive! Authors should reduce unuseful descriptions. Because of this, soem readers loose interest for this interesting subject;

Response: The authors appreciate the reviewer’s comment, however, they believe that the definitions of several terms (e.g., cyber-attack path) ought to be descriptive enough for readers coming from various fields to fully understand the content of the paper.  Additionally, we have carefully further revised the paper to address any unuseful description.

Comment: Paper is too analytic; some more real applications must be presented for validation of the presented approach for "Cyber Attack Path Generation and Prioritisation for Securing the Healthcare Systems";

Response: We have implemented a real healthcare use case scenario based on a user-centred Digital Health Living lab. This is a Tier 3 test and trial category according to the UK National Institute for Health and Care Excellence (NICE) for Digital Health Technologies (DHTs). The proposed approach is systematically implemented in terms of identification of entry point and target point assets for the possible attack path. Our result prioritises a number of attack paths based on the security incident and vulnerability  evidence chain. Please note that this real application validates the proposed work and  demonstrates our finding to achieve a secure healthcare service delivery.

The main added value of this work - when compared with other similar ones - is not clearly presnted to the readers! Authors must be more objective in their sentences and explanations;

Response:   We have reviewed the works related with the proposed approach. In particular, there are a limited number of recent works that focus on attack path discovery mainly critical infrastructure and dynamic maritime supply chain context. These works made some important contributions for attack path generation. But, there is a lack of focus on vulnerability exploitability and target user profiles for the attack path. This is important in the healthcare sector due to the dependencies of IT and healthcare devices and lack of healthcare practitioners awareness about security. Our work contributes towards this direction. The unique contributions of this work is added at the end of introduction (section 1),  related work (section 2) and attack path discovery method (section 3)

Conclusions must be improved and more objective about the main added value of the paper.

If some real improvements will be performed, perhaps it could be an interesting paper, due to the subject, very important nowadays. 

Response:  The conclusions have now been modified to be more targeted and cohesive. In particular, the benefits of the proposed approach and its implementation are discussed.

Round 2

Reviewer 1 Report

The whole paper has been greatly improved, but in subsection 5.3, only 5.3.1, without 5.3.2, the structure of the paper is inappropriate. Careful over and over revision of the paper is recommended to meet publication standards.

Author Response

Authors’ response to Reviewers’ Comments

Applied science

MDPI

Manuscript ID: applsci-1635858

Title: Cyber Attack Path Generation and Prioritisation for Securing the Healthcare Systems

We want to thank the reviewers for their time and effort to review our  revised paper and for their comments. To facilitate a more straightforward reading of our responses, we have used the following format in the text below:

  • Each comment has been highlighted in bold.
  • After each reviewer’s comment, our corresponding reply is provided.

Comments: The whole paper has been greatly improved, but in subsection 5.3, only 5.3.1, without 5.3.2, the structure of the paper is inappropriate. Careful over and over revision of the paper is recommended to meet publication standards.

Response: Thank you for this comment.  By following this comment, the subsection number 5.3.1 is removed. The whole paper is further reviewed to address any issue. We believe the current version will meet the publication standard.

Reviewer 2 Report

All issues have been, somehow, answered by authors.

I recommend the publication of the paper.

Author Response

Authors’ response to Reviewers’ Comments

Applied science

MDPI

Manuscript ID: applsci-1635858

Title: Cyber Attack Path Generation and Prioritisation for Securing the Healthcare Systems

We want to thank the reviewers for their time and effort to review our  revised paper and for their comments. To facilitate a more straightforward reading of our responses, we have used the following format in the text below:

  • Each comment has been highlighted in bold.
  • After each reviewer’s comment, our corresponding reply is provided.

Comment : All issues have been, somehow, answered by authors.

I recommend the publication of the paper.

Response: Thank you for this comment and accept our paper. We have also revised the whole manuscript to address any issue.